# OFFLINE REINFORCEMENT LEARNING WITH IMPLICIT Q-LEARNING

**Ilya Kostrikov, Ashvin Nair & Sergey Levine**
Department of Electrical Engineering and Computer Science
University of California, Berkeley
`{kostrikov,anair17}@berkeley.edu, svlevine@eecs.berkeley.edu`

## ABSTRACT

Offline reinforcement learning requires reconciling two conflicting aims: learning a policy that improves over the behavior policy that collected the dataset, while at the same time minimizing the deviation from the behavior policy so as to avoid errors due to distributional shift. This trade-off is critical, because most current offline reinforcement learning methods need to query the value of unseen actions during training to improve the policy, and therefore need to either constrain these actions to be in-distribution, or else regularize their values. We propose a new offline RL method that never needs to evaluate actions outside of the dataset, but still enables the learned policy to improve substantially over the best behavior in the data through generalization. The main insight in our work is that, instead of evaluating unseen actions from the latest policy, we can approximate the policy improvement step implicitly by treating the state value function as a random variable, with randomness determined by the action (while still integrating over the dynamics to avoid excessive optimism), and then taking a state conditional upper expectile of this random variable to estimate the value of the best actions in that state. This leverages the generalization capacity of the function approximator to estimate the value of the best available action at a given state without ever directly querying a Q-function with this unseen action. Our algorithm alternates between fitting this upper expectile value function and backing it up into a Q-function, without any explicit policy. Then, we extract the policy via advantage-weighted behavioral cloning, which also avoids querying out-of-sample actions. We dub our method implicit Q-learning (IQL). IQL is easy to implement, computationally efficient, and only requires fitting an additional critic with an asymmetric L2 loss. IQL demonstrates the state-of-the-art performance on D4RL, a standard benchmark for offline reinforcement learning. We also demonstrate that IQL achieves strong performance fine-tuning using online interaction after offline initialization.

## 1 INTRODUCTION

Offline reinforcement learning (RL) addresses the problem of learning effective policies entirely from previously collected data, without online interaction (Fujimoto et al., 2019; Lange et al., 2012). This is very appealing in a range of real-world domains, from robotics to logistics and operations research, where real-world exploration with untrained policies is costly or dangerous, but prior data is available. However, this also carries with it major challenges: improving the policy beyond the level of the behavior policy that collected the data requires estimating values for actions other than those that were seen in the dataset, and this, in turn, requires trading off policy improvement against distributional shift, since the values of actions that are too different from those in the data are unlikely to be estimated accurately. Prior methods generally address this by either constraining the policy to limit how far it deviates from the behavior policy (Fujimoto et al., 2019; Wu et al., 2019; Fujimoto & Gu, 2021; Kumar et al., 2019; Nair et al., 2020; Wang et al., 2020), or by regularizing the learned value functions to assign low values to out-of-distribution actions (Kumar et al., 2020; Kostrikov et al., 2021). Nevertheless, this imposes a trade-off between how much the policy improves and how vulnerable it is to misestimation due to distributional shift. Can we devise an offline RL method that

avoids this issue by never needing to directly query or estimate values for actions that were not seen in the data?

In this work, we start from an observation that *in-distribution* constraints widely used in prior work might not be sufficient to avoid value function extrapolation, and we ask whether it is possible to learn an optimal policy with *in-sample learning*, without *ever* querying the values of any unseen actions. The key idea in our method is to approximate an upper expectile of the distribution over values with respect to the distribution of dataset actions for each state. We alternate between fitting this value function with expectile regression, and then using it to compute Bellman backups for training the $Q$-function. We show that we can do this simply by modifying the loss function in a SARSA-style TD backup, without ever using out-of-sample actions in the target value. Once this $Q$-function has converged, we extract the corresponding policy using advantage-weighted behavioral cloning. This approach does not require explicit constraints or explicit regularization of out-of-distribution actions during value function training, though our policy extraction step does implicitly enforce a constraint, as discussed in prior work on advantage-weighted regression (Peters & Schaal, 2007; Peng et al., 2019; Nair et al., 2020; Wang et al., 2020).

Our main contribution is implicit Q-learning (IQL), a new offline RL algorithm that avoids ever querying values of unseen actions while still being able to perform multi-step dynamic programming updates. Our method is easy to implement by making a small change to the loss function in a simple SARSA-like TD update and is computationally very efficient. Furthermore, our approach demonstrates the state-of-the-art performance on D4RL, a popular benchmark for offline reinforcement learning. In particular, our approach significantly improves over the prior state-of-the-art on challenging Ant Maze tasks that require to "stitch" several sub-optimal trajectories. Finally, we demonstrate that our approach is suitable for finetuning; after initialization from offline RL, IQL is capable of improving policy performance utilizing additional interactions.

## 2 RELATED WORK

A significant portion of recently proposed offline RL methods are based on either constrained or regularized approximate dynamic programming (e.g., Q-learning or actor-critic methods), with the constraint or regularizer serving to limit deviation from the behavior policy. We will refer to these methods as "multi-step dynamic programming" algorithms, since they perform true dynamic programming for multiple iterations, and therefore can in principle recover the optimal policy if provided with high-coverage data. The constraints can be implemented via an explicit density model (Wu et al., 2019; Fujimoto et al., 2019; Kumar et al., 2019; Ghasemipour et al., 2021), implicit divergence constraints (Nair et al., 2020; Wang et al., 2020; Peters & Schaal, 2007; Peng et al., 2019; Siegel et al., 2020), or by adding a supervised learning term to the policy improvement objective (Fujimoto & Gu, 2021). Several works have also proposed to directly regularize the Q-function to produce low values for out-of-distribution actions (Kostrikov et al., 2021; Kumar et al., 2020; Fakoor et al., 2021). Our method is also a multi-step dynamic programming algorithm. However, in contrast to prior works, our method completely avoids directly querying the learned Q-function with unseen actions during training, removing the need for any constraint during this stage, though the subsequent policy extraction, which is based on advantage-weighted regression (Peng et al., 2019; Nair et al., 2020), does apply an implicit constraint. However, this policy does not actually influence value function training.

In contrast to multi-step dynamic programming methods, several recent works have proposed methods that rely either on a single step of policy iteration, fitting the value function or Q-function of the behavior policy and then extracting the corresponding greedy policy (Peng et al., 2019; Brandfonbrener et al., 2021; Gulcehre et al., 2021), or else avoid value functions completely and utilize behavioral cloning-style objectives (Chen et al., 2021). We collectively refer to these as "single-step" approaches. These methods avoid needing to query unseen actions as well, since they either use no value function at all, or learn the value function of the behavior policy. Although these methods are simple to implement and effective on the MuJoCo locomotion tasks in D4RL, we show that such single-step methods perform very poorly on more complex datasets in D4RL, which require combining parts of suboptimal trajectories ("stitching"). Prior multi-step dynamic programming methods perform much better in such settings, as does our method. We discuss this distinction in more detail in Section 5.1. Our method also shares the simplicity and computational efficiency of single-step approaches, providing an appealing combination of the strengths of both types of methods.

Our method is based on estimating the characteristics of a random variable. Several recent works involve approximating statistical quantities of the value function distribution. In particular, quantile regression (Koenker & Hallock, 2001) has been previously used in reinforcement learning to estimate the quantile function of a state-action value function (Dabney et al., 2018b;a; Kuznetsov et al., 2020). Although our method is related, in that we perform expectile regression, our aim is not to estimate the distribution of values that results from stochastic transitions, but rather estimate expectiles of the state value function with respect to random actions. This is a very different statistic: our aim is not to determine how the $Q$-value can vary with different future outcomes, but how the $Q$-value can vary with different actions *while averaging together future outcomes due to stochastic dynamics*. While prior work on distributional RL can also be used for offline RL, it would suffer from the same action extrapolation issues as other methods, and would require similar constraints or regularization, while our method does not.

## 3  PRELIMINARIES

The RL problem is formulated in the context of a Markov decision process (MDP) $(\mathcal{S}, \mathcal{A}, p_0(s), p(s'|s, a), r(s, a), \gamma)$, where $\mathcal{S}$ is a state space, $\mathcal{A}$ is an action space, $p_0(s)$ is a distribution of initial states, $p(s'|s, a)$ is the environment dynamics, $r(s, a)$ is a reward function, and $\gamma$ is a discount factor. The agent interacts with the MDP according to a policy $\pi(a|s)$. The goal is to obtain a policy that maximizes the cumulative discounted returns:

$$\pi^* = \arg\max_{\pi} \mathbb{E}_{\pi}\left[\sum_{t=0}^{\infty} \gamma^t r(s_t, a_t)|s_0 \sim p_0(\cdot), a_t \sim \pi(\cdot|s_t), s_{t+1} \sim p(\cdot|s_t, a_t)\right].$$

Off-policy RL methods based on approximate dynamic programming typically utilize a state-action value function ($Q$-function), referred to as $Q(s, a)$, which corresponds to the discounted returns obtained by starting from the state $s$ and action $a$, and then following the policy $\pi$.

**Offline reinforcement learning.**  In contrast to online (on-policy or off-policy) RL methods, offline RL uses previously collected data without any additional data collection. Like many recent offline RL methods, our work builds on approximate dynamic programming methods that minimize temporal difference error, according to the following loss:

$$L_{TD}(\theta) = \mathbb{E}_{(s,a,s')\sim\mathcal{D}}[(r(s, a) + \gamma \max_{a'} Q_{\hat{\theta}}(s', a') - Q_{\theta}(s, a))^2], \tag{1}$$

where $\mathcal{D}$ is the dataset, $Q_{\theta}(s, a)$ is a parameterized $Q$-function, $Q_{\hat{\theta}}(s, a)$ is a target network (e.g., with soft parameters updates defined via Polyak averaging), and the policy is defined as $\pi(s) = \arg\max_a Q_{\theta}(s, a)$. Most recent offline RL methods modify either the value function loss (above) to regularize the value function in a way that keeps the resulting policy close to the data, or constrain the $\arg\max$ policy directly. This is important because out-of-distribution actions $a'$ can produce erroneous values for $Q_{\hat{\theta}}(s', a')$ in the above objective, often leading to *overestimation* as the policy is defined to maximize the (estimated) Q-value.

## 4  IMPLICIT Q-LEARNING

In this work, we aim to entirely avoid querying *out-of-sample* (unseen) actions in our TD loss. Although the goal of this work is to approximate the optimal $Q$-function, we start by considering fitted $Q$ evaluation with a SARSA-style objective which has been considered in prior work on Offline Reinforcement Learning (Brandfonbrener et al., 2021; Gulcehre et al., 2021). This objective aims to learn the value of the dataset policy $\pi_{\beta}$ (also called the behavior policy):

$$L(\theta) = \mathbb{E}_{(s,a,s',a')\sim\mathcal{D}}[(r(s, a) + \gamma Q_{\hat{\theta}}(s', a') - Q_{\theta}(s, a))^2]. \tag{2}$$

This objective never queries values for out-of-sample actions, in contrast to Eqn. (1). One specific property of this objective that is important for this work is that it uses mean squared error (MSE) that fits $Q_{\theta}(s, a)$ to predict the mean statistics of the TD targets. Thus, if we assume unlimited capacity and no sampling error, the optimal parameters should satisfy

$$Q_{\theta^*}(s, a) \approx r(s, a) + \gamma \mathbb{E}_{\substack{s' \sim p(\cdot|s,a) \\ a' \sim \pi_{\beta}(\cdot|s)}}[Q_{\hat{\theta}}(s', a')]. \tag{3}$$

Prior work (Brandfonbrener et al., 2021; Gulcehre et al., 2021; Peng et al., 2019) has proposed directly using this objective to learn $Q^{\pi_{\beta}}$, and then train the policy $\pi_{\psi}$ to maximize $Q^{\pi_{\beta}}$. This avoids

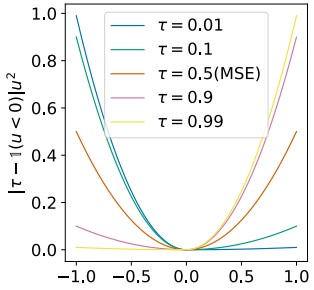 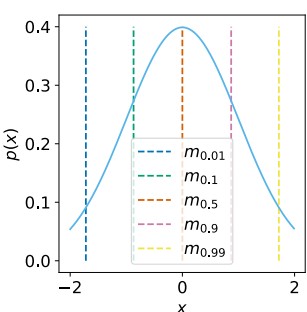 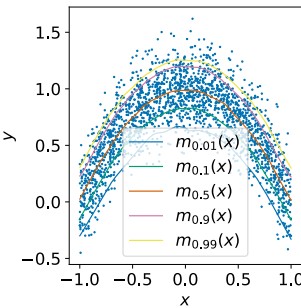

Figure 1: **Left:** The asymmetric squared loss used for expectile regression. $\tau = 0.5$ corresponds to the standard mean squared error loss, while $\tau = 0.9$ gives more weight to positives differences. **Center:** Expectiles of a normal distribution. **Right:** an example of estimating state conditional expectiles of a two-dimensional random variable. Each $x$ corresponds to a distribution over $y$. We can approximate a maximum of this random variable with expectile regression: $\tau = 0.5$ correspond to the conditional mean statistics of the distribution, while $\tau \approx 1$ approximates the maximum operator over in-support values of $y$.

any issues with out-of-distribution actions, since the TD loss only uses dataset actions. However, while this procedure works well empirically on simple MuJoCo locomotion tasks in D4RL, we will show that it performs very poorly on more complex tasks that benefit from multi-step dynamic programming. In our method, which we derive next, we retain the benefits of using this SARSA-like objective, but modify it so that it allows us to perform multi-step dynamic programming and learn a near-optimal Q-function.

Our method will perform a $Q$-function update similar to Eqn. (2), but we will aim to estimate the maximum $Q$-value over actions that are in the support of the data distribution. Crucially, we will show that it is possible to do this *without ever querying the learned Q-function on out-of-sample actions* by utilizing expectile regression. Formally, the value function we aim to learn is given by:

$$L(\theta) = \mathbb{E}_{(s,a,s')\sim\mathcal{D}}[(r(s,a) + \gamma \max_{\substack{a' \in \mathcal{A} \\ \text{s.t. } \pi_\beta(a'|s')>0}} Q_{\hat{\theta}}(s',a') - Q_\theta(s,a))^2]. \quad (4)$$

Our algorithm, implicit Q-Learning (IQL), aims to estimate this objective while evaluating the $Q$-function only on the state-action pairs in the dataset. To this end, we propose to fit $Q_\theta(s,a)$ to estimate state-conditional expectiles of the target values, and show that specific expectiles approximate the maximization defined above. In Section 4.4 we show that this approach performs multi-step dynamic programming in theory, and in Section 5.1 we show that it does so in practice.

### 4.1 EXPECTILE REGRESSION

Practical methods for estimating various statistics of a random variable have been thoroughly studies in applied statistics and econometrics. The $\tau \in (0,1)$ expectile of some random variable $X$ is defined as a solution to the asymmetric least squares problem:

$$\arg\min_{m_\tau} \mathbb{E}_{x\sim X}[L_2^\tau(x - m_\tau)], \text{ where } L_2^\tau(u) = |\tau - \mathbb{1}(u < 0)|u^2.$$

That is, for $\tau > 0.5$, this asymmetric loss function downweights the contributions of $x$ values smaller than $m_\tau$ while giving more weights to larger values (see Fig. 1, left). Expectile regression is closely related to quantile regression (Koenker & Hallock, 2001), which is a popular technique for estimating quantiles of a distribution widely used in reinforcement learning (Dabney et al., 2018b;a) [1]. The quantile regression loss is defined as an asymmetric $\ell_1$ loss.

---

[1]Our method could also be derived with quantiles, but since we are not interested in learning all of the expectiles/quantiles, unlike prior work (Dabney et al., 2018b;a), it is more convenient to estimate a single expectile because this involves a simple modification to the MSE loss that is already used in standard RL methods. We found it to work somewhat better than quantile regression with its corresponding $\ell_1$ loss.

We can also use this formulation to predict expectiles of a conditional distribution:

$$\underset{m_\tau(x)}{\arg\min} \, \mathbb{E}_{(x,y)\sim\mathcal{D}}[L_2^\tau(y - m_\tau(x))].$$

Fig. 1 (right) illustrates conditional expectile regression on a simple two-dimensional distribution. Note that we can optimize this objective with stochastic gradient descent. It provides unbiased gradients and is easy to implement with standard machine learning libraries.

## 4.2 LEARNING THE VALUE FUNCTION WITH EXPECTILE REGRESSION

Expectile regression provides us with a powerful framework to estimate statistics of a random variable beyond mean regression. We can use expectile regression to modify the policy evaluation objective in Eqn. (2) to predict an upper expectile of the TD targets that approximates the maximum of $r(s,a) + \gamma Q_{\hat\theta}(s', a')$ over actions $a'$ constrained to the dataset actions, as in Eqn. (4). This leads to the following expectile regression objective:

$$L(\theta) = \mathbb{E}_{(s,a,s',a')\sim\mathcal{D}}[L_2^\tau(r(s,a) + \gamma Q_{\hat\theta}(s', a') - Q_\theta(s,a))].$$

However, this formulation has a significant drawback. Instead of estimating expectiles just with respect to the actions in the support of the data, it also incorporates stochasticity that comes from the environment dynamics $s' \sim p(\cdot|s,a)$. Therefore, a large target value might not necessarily reflect the existence of a single action that achieves that value, but rather a "lucky" sample that happened to have transitioned into a good state. We resolve this by introducing a separate value function that approximates an expectile only with respect to the action distribution, leading to the following loss:

$$L_V(\psi) = \mathbb{E}_{(s,a)\sim\mathcal{D}}[L_2^\tau(Q_{\hat\theta}(s,a) - V_\psi(s))]. \tag{5}$$

We can then use this estimate to update the $Q$-functions with the MSE loss, which averages over the stochasticity from the transitions and avoids the "lucky" sample issue mentioned above:

$$L_Q(\theta) = \mathbb{E}_{(s,a,s')\sim\mathcal{D}}[(r(s,a) + \gamma V_\psi(s') - Q_\theta(s,a))^2]. \tag{6}$$

Note that these losses do not use any explicit policy, and only utilize actions from the dataset for both objectives, similarly to SARSA-style policy evaluation. In Section 4.4, we will show that this procedure recovers the optimal Q-function under some assumptions. Also, even though only one action is available for every state in the dataset for continuous action spaces, due to neural network generalization, the expectile regression does not result in SARSA-style policy evaluation as shown in Section 5.2.

## 4.3 POLICY EXTRACTION AND ALGORITHM SUMMARY

While our modified TD learning procedure learns an approximation to the optimal Q-function, it does not explicitly represent the corresponding policy, and therefore requires a separate policy extraction step. While one can consider any technique for policy extraction that constrains the learned policy to stay close to the dataset actions, we aim for a simple method for policy extraction. As before, we aim to avoid using out-of-samples actions. Therefore, we extract the policy with advantage-weighted regression (Peters & Schaal, 2007; Peng et al., 2019) previously successfully used for policy extraction in Offline RL (Wang et al., 2018; Nair et al., 2020; Brandfonbrener et al., 2021):

$$L_\pi(\phi) = \mathbb{E}_{(s,a)\sim\mathcal{D}}[\exp(\beta(Q_{\hat\theta}(s,a) - V_\psi(s)))\log\pi_\phi(a|s)], \tag{7}$$

where $\beta \in [0, \infty)$ is an inverse temperature. Note that this objective does not clone all actions from the dataset but, as

---

**Algorithm 1** Implicit Q-learning

Initialize parameters $\psi, \theta, \hat\theta, \phi$.
TD learning (IQL):
**for** each gradient step **do**
    $\psi \leftarrow \psi - \lambda_V \nabla_\psi L_V(\psi)$
    $\theta \leftarrow \theta - \lambda_Q \nabla_\theta L_Q(\theta)$
    $\hat\theta \leftarrow (1 - \alpha)\hat\theta + \alpha\theta$
**end for**
Policy extraction (AWR):
**for** each gradient step **do**
    $\phi \leftarrow \phi - \lambda_\pi \nabla_\phi L_\pi(\phi)$
**end for**

---

shown in prior work, this objective learns a policy that maximizes the $Q$-values subject to a distribution constraint (Peters & Schaal, 2007; Peng et al., 2019; Nair et al., 2020). This step can be seen as selecting and cloning the most optimal actions in the dataset.

Our final algorithm consists of two stages. First, we fit the value function and $Q$, performing a number of gradient updates alternating between Eqn. (5) and (6). Second, we perform stochastic gradient descent on Eqn. (7). For both steps, we use a version of clipped double $Q$-learning (Fujimoto et al., 2018), taking a minimum of two $Q$-functions for $V$-function and policy updates. We summarize our final method in Algorithm 1. Note that the policy does not influence the value function in any way, and therefore extraction could be performed either concurrently or after TD learning. Concurrent learning provides a way to use IQL with online finetuning, as we discuss in Section 5.3.

## 4.4 ANALYSIS

In this section, we will show that IQL can recover the optimal value function under the dataset support constraints. First, we prove a simple lemma that we will then use to show how our approach can enable learning the optimal value function.

**Lemma 1.** *Let $X$ be a real-valued random variable with a bounded support and supremum of the support is $x^*$. Then,*

$$\lim_{\tau \to 1} m_\tau = x^*$$

*Proof Sketch.* One can show that expectiles of a random variable have the same supremum $x^*$. Moreover, for all $\tau_1$ and $\tau_2$ such that $\tau_1 < \tau_2$, we get $m_{\tau_1} \le m_{\tau_2}$. Therefore, the limit follows from the properties of bounded monotonically non-decreasing functions. $\square$

In the following theorems, we show that under certain assumptions, our method indeed approximates the optimal state-action value $Q^*$ and performs multi-step dynamical programming. We first prove a technical lemma relating different expectiles of the $Q$-function, and then derive our main result regarding the optimality of our method.

For the sake of simplicity, we introduce the following notation for our analysis. Let $\mathbb{E}^\tau_{x \sim X}[x]$ be a $\tau^{\text{th}}$ expectile of $X$ (e.g., $\mathbb{E}^{0.5}$ corresponds to the standard expectation). Then, we define $V_\tau(s)$ and $Q_\tau(s, a)$, which correspond to optimal solutions of Eqn. 5 and 6 correspondingly, recursively as:

$$V_\tau(s) = \mathbb{E}^\tau_{a \sim \pi_\beta(\cdot|s)}[Q_\tau(s, a)],$$

$$Q_\tau(s, a) = r(s, a) + \gamma \mathbb{E}_{s' \sim p(\cdot|s,a)}[V_\tau(s')].$$

**Lemma 2.** *For all $s$, $\tau_1$ and $\tau_2$ such that $\tau_1 < \tau_2$ we get $V_{\tau_1}(s) \le V_{\tau_2}(s)$.*

*Proof.* The proof follows the policy improvement proof (Sutton & Barto, 2018). See Appendix A. $\square$

**Corollary 2.1.** *For any $\tau$ and $s$ we have $V_\tau(s) \le \max\limits_{\substack{a \in \mathcal{A} \\ s.t. \ \pi_\beta(a|s) > 0}} Q^*(s, a)$ where $V_\tau(s)$ is defined as above and $Q^*(s, a)$ is an optimal state-action value function constrained to the dataset and defined as*

$$Q^*(s, a) = r(s, a) + \gamma \mathbb{E}_{s' \sim p(\cdot|s,a)} \left[ \max\limits_{\substack{a' \in \mathcal{A} \\ s.t. \ \pi_\beta(a'|s') > 0}} Q^*(s', a') \right].$$

*Proof.* The proof follows from the observation that convex combination is smaller than maximum. $\square$

**Theorem 3.**

$$\lim_{\tau \to 1} V_\tau(s) = \max\limits_{\substack{a \in \mathcal{A} \\ s.t. \ \pi_\beta(a|s) > 0}} Q^*(s, a).$$

*Proof.* Follows from combining Lemma 1 and Corollary 2.1. $\square$

Therefore, for a larger value of $\tau < 1$, we get a better approximation of the maximum. On the other hand, it also becomes a more challenging optimization problem. Thus, we treat $\tau$ as a hyperparameter. Due to the property discussed in Theorem 3 we dub our method implicit $Q$-learning (IQL). We also emphasize that our value learning method defines the entire spectrum of methods between SARSA ($\tau = 0.5$) and Q-Learning ($\tau \to 1$). Note that in contrast to other multi-step methods, IQL absorbs the policy improvement step into value learning. Therefore, fitting Q-function corresponds to the policy evaluation step, while fitting the value function with IQL corresponds to *implicit* policy improvement.

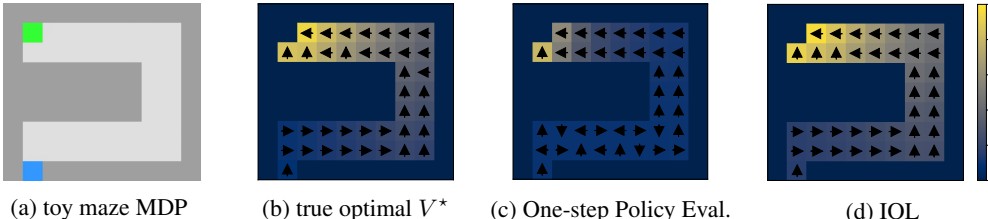

(a) toy maze MDP          (b) true optimal $V^\star$          (c) One-step Policy Eval.          (d) IQL

Figure 2: Evaluation of our algorithm on a toy umaze environment (a). When the static dataset is heavily corrupted by suboptimal actions, one-step policy evaluation results in a value function that degrades to zero far from the rewarding states too quickly (c). Our algorithm aims to learn a near-optimal value function, combining the best properties of SARSA-style evaluation with the ability to perform multi-step dynamic programming, leading to value functions that are much closer to optimality (shown in (b)) and producing a much better policy (d).

## 5 EXPERIMENTAL EVALUATION

Our experiments aim to evaluate our method comparatively, in contrast to prior offline RL methods, and in particular to understand how our approach compares both to single-step methods and multi-step dynamic programming approaches. We will first demonstrate the benefits of multi-step dynamic programming methods, such as ours, in contrast to single-step methods, showing that on some problems this difference can be extremely large. We will then compare IQL with state-of-the-art single-step and multi-step algorithms on the D4RL (Fu et al., 2020) benchmark tasks, studying the degree to which we can learn effective policies using only the actions in the dataset. We examine domains that contain near-optimal trajectories, where single-step methods perform well, as well as domains with no optimal trajectories at all, which require multi-step dynamic programming. Finally, we will study how IQL compares to prior methods when finetuning with online RL starting from an offline RL initialization.

### 5.1 THE DIFFERENCE BETWEEN ONE-STEP POLICY IMPROVEMENT AND IQL

We first use a simple maze environment to illustrate the importance of multi-step dynamic programming for offline RL. The maze has a u-shape, a single start state, and a single goal state (see Fig. 2a). The agent receives a reward of 10 for entering the goal state and zero reward for all other transitions. With a probability of 0.25, the agent transitions to a random state, and otherwise to the commanded state. The dataset consists of 1 optimal trajectory and 99 trajectories with uniform random actions. Due to a short horizon of the problem, we use $\gamma = 0.9$.

Fig. 2 (c, d) illustrates the difference between single-step methods which fit $Q^\pi(s, a)$ via SARSA-style objective, in this case represented by Onepstep RL (Brandfonbrener et al., 2021; Wang et al., 2018; Gulcehre et al., 2021) and IQL with $\tau = 0.95$. Note that these methods represent a special case of our method with $\tau = 0.5$. Although states closer to the high reward state will still have higher values, these values decay much faster as we move further away than they would for the optimal value function, and the resulting policy is highly suboptimal. Since IQL (d) performs iterative dynamic programming, it correctly propagates the signal, and the values are no longer dominated by noise. The resulting value function closely matches the true optimal value function (b).

### 5.2 COMPARISONS ON OFFLINE RL BENCHMARKS

Next, we evaluate our approach on the D4RL benchmark in comparison to prior methods (see Table 1). The MuJoCo tasks in D4RL consist of the Gym locomotion tasks, the Ant Maze tasks, and the Adroit and Kitchen robotic manipulation environments. Some prior works, particularly those proposing one-step methods, focus entirely on the Gym locomotion tasks. However, these tasks include a significant fraction of near-optimal trajectories in the dataset. In contrast, the Ant Maze tasks, especially the medium and large ones, contain very few or no near-optimal trajectories, making them very challenging for one-step methods. These domains require "stitching" parts of suboptimal trajectories that travel between different states to find a path from the start to the goal of the maze (Fu et al., 2020). As we will show, multi-step dynamic programming is essential in these domains. The Adroit and Kitchen tasks are comparatively less discriminating, and we found that most RL methods perform similarly to imitation learning in these domains (Florence et al., 2021).

Table 1: Averaged normalized scores on MuJoCo locomotion and Ant Maze tasks. Our method outperforms prior methods on the challenging Ant Maze tasks, which require dynamic programming, and is competitive with the best prior methods on the locomotion tasks.

| Dataset | BC | 10%BC | BCQ | DT | ABM | AWAC | Onestep RL | TD3+BC | CQL | IQL (Ours) |
|---|---|---|---|---|---|---|---|---|---|---|
| halfcheetah-m-v2 | 42.6 | 42.5 | **47.0** | 42.6±0.1 | 53.6 | 43.5 | **48.4±0.1** | 48.3±0.3 | 44.0±5.4 | 47.4±0.2 |
| hopper-m-v2 | 52.9 | 56.9 | 56.7 | **67.6±1.0** | 0.7 | 57.0 | 59.6±2.5 | 59.3±4.2 | 58.5±2.1 | **66.2±5.7** |
| walker2d-m-v2 | 75.3 | 75.0 | 72.6 | 74.0±1.4 | 0.5 | 72.4 | **81.8±2.2** | 83.7±2.1 | 72.5±0.8 | 78.3± 8.7 |
| halfcheetah-m-r-v2 | 36.6 | 40.6 | 40.4 | 36.6±0.8 | **50.5** | 40.5 | 38.1±1.3 | **44.6±0.5** | 45.5±0.5 | 44.2±1.2 |
| hopper-m-r-v2 | 18.1 | 75.9 | 53.3 | 82.7±7.0 | 49.6 | 37.2 | **97.5±0.7** | 60.9±18.8 | **95.0±6.4** | 94.7±8.6 |
| walker2d-m-r-v2 | 26.0 | 62.5 | 52.1 | 66.6±3.0 | 53.8 | 27.0 | 49.5±12.0 | **81.8±5.5** | 77.2±5.5 | 73.8±7.1 |
| halfcheetah-m-e-v2 | 55.2 | **92.9** | 89.1 | 86.8±1.3 | 18.5 | 42.8 | **93.4±1.6** | 90.7±4.3 | 91.6±2.8 | 86.7±5.3 |
| hopper-m-e-v2 | 52.5 | **110.9** | 81.8 | **107.6±1.8** | 0.7 | 55.8 | 103.3±1.9 | 98.0±9.4 | 105.4±6.8 | 91.5±14.3 |
| walker2d-m-e-v2 | 107.5 | 109.0 | 109.5 | 108.1±0.2 | 3.5 | 74.5 | **113.0±0.4** | 110.1±0.5 | 108.8±0.7 | 109.6±1.0 |
| locomotion-v2 total | 466.7 | **666.2** | 602.5 | 672.6±16.6 | 231.4 | 450.7 | **684.6±22.7** | 677.4±44.5 | **698.5±31.0** | 692.4±52.1 |
| antmaze-u-v0 | 54.6 | 62.8 | **89.8** | 59.2 | 59.9 | 56.7 | 64.3 | 78.6 | 74.0 | **87.5 ± 2.6** |
| antmaze-u-d-v0 | 45.6 | 50.2 | **83.0** | 53.0 | 48.7 | 49.3 | 60.7 | 71.4 | **84.0** | 62.2 ± 13.8 |
| antmaze-m-p-v0 | 0.0 | 5.4 | 15.0 | 0.0 | 0.0 | 0.0 | 0.3 | 10.6 | 61.2 | **71.2 ± 7.3** |
| antmaze-m-d-v0 | 0.0 | 9.8 | 0.0 | 0.0 | 0.5 | 0.7 | 0.0 | 3.0 | 53.7 | **70.0 ± 10.9** |
| antmaze-l-p-v0 | 0.0 | 0.0 | 0.0 | 0.0 | 0. | 0.0 | 0.0 | 0.2 | 15.8 | **39.6±5.8** |
| antmaze-l-d-v0 | 0.0 | 6.0 | 0.0 | 0.0 | 0.0 | 1.0 | 0.0 | 0.0 | 14.9 | **47.5±9.5** |
| antmaze-v0 total | 100.2 | 134.2 | 187.8 | 112.2 | 109.1 | 107.7 | 125.3 | 163.8 | 303.6 | **378.0±49.9** |
| total | 566.9 | 800.4 | 790.3 | 784.8 | 340.5 | 558.4 | 809.9 | 841.2 | 1002.1 | **1070.4±102.0** |
| kitchen-v0 total | **154.5** | - | - | - | - | - | - | - | 144.6 | **159.8±22.6** |
| adroit-v0 total | 104.5 | - | - | - | - | - | - | - | 93.6 | **118.1±30.7** |
| total+kitchen+adroit | 825.9 | - | - | - | - | - | - | - | 1240.3 | **1348.3±155.3** |
| runtime | 10m | 10m | | 960m | | 20m | 20m* | 20m | 80m | 20m |

*: Note that it is challenging to compare one-step and multi-step methods directly. Also, Brandfonbrener et al. (2021) reports results for a set of hyperparameters, such as batch and network size, that is significantly different from other methods. We report results for the original hyperparameters and runtime for a comparable set of hyperparameters.

We therefore focus our analysis on the Gym locomotion and Ant Maze domains, but include full Adroit and Kitchen results in Appendix B for completeness.

**Comparisons and baselines.** We compare to methods that are representative of both multi-step dynamic programming and one-step approaches. In the former category, we compare to CQL (Kumar et al., 2020), TD3+BC (Fujimoto & Gu, 2021), and AWAC (Nair et al., 2020). In the latter category, we compare to Onestep RL (Brandfonbrener et al., 2021) and Decision Transformers (Chen et al., 2021). We obtained the Decision Transformers results on Ant Maze subsets of D4RL tasks using the author-provided implementation[2] and following authors instructions communicated over email. We obtained results for TD3+BC and Onestep RL (Exp. Weight) directly from the authors. Note that Chen et al. (2021) and Brandfonbrener et al. (2021) incorrectly report results for some prior methods, such as CQL, using the "-v0" environments. These generally produce lower scores than the "-v2" environments that these papers use for their own methods. We use the "-v2" environments for all methods to en-

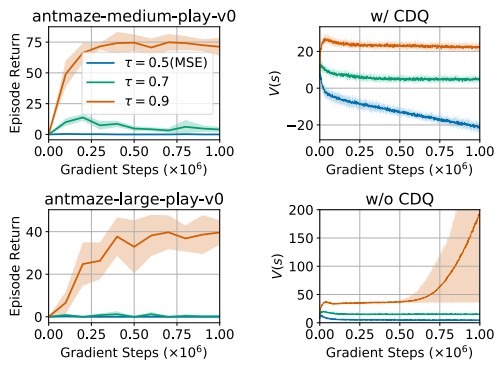

Figure 3: **Left**: Estimating a larger expectile $\tau$ is crucial for antmaze tasks that require dynamical programming ('stitching'). **Right:** Clipped double Q-Learning (CDQ) is crucial for learning values for $\tau = 0.9$.

sure a fair comparison, resulting in higher values for CQL. Because of this fix, our reported CQL scores are higher than all other prior methods. We obtained results for "-v2" datasets using an author-suggested implementation.[3] On the Gym locomotion tasks (halfcheetah, hopper, walker2d), we find that IQL performs comparably to the best performing prior method, CQL. On the more challenging Ant Maze task, IQL outperforms CQL, and outperforms the one-step methods by a very large margin.

**Runtime.** Our approach is also computationally faster than the baselines (see Table 1). For the baselines, we measure runtime for our reimplementations of the methods in JAX (Bradbury et al., 2018) built on top of JAXRL (Kostrikov, 2021), which are typically faster than the original imple-

---

[2]https://github.com/kzl/decision-transformer
[3]https://github.com/young-geng/CQL

mentations. For example, the original implementation of CQL takes more than 4 hours to perform 1M updates, while ours takes only 80 minutes. Even so, IQL still requires about 4x less time than our reimplementation of CQL on average, and is comparable to the fastest prior one-step methods. We did not reimplement Decision Transformers due to their complexity and report runtime of the original implementation.

**Effect of $\tau$ hyperparameter.** We also demonstrate that it is crucial to compute a larger expectile on tasks that require "stitching" (see Fig. 3). We provide complete results in Appendix B. With larger values of $\tau$, our method approximates $Q$-learning better, leading to better performance on the Ant Maze tasks. Moreover, due to neural network generalization, values learned with expectile regression increase with a larger $\tau$ and do not degrade to behavior policy values ($\tau = 0.5$). Finally, clipped double Q-Learning is crucial for estimating values for a larger $\tau = 0.9$.

### 5.3 ONLINE FINE-TUNING AFTER OFFLINE RL

The policies obtained by offline RL can often be improved with a small amount of online interaction. IQL is well-suited for online fine-tuning for two reasons. First, IQL has strong offline performance, as shown in the previous section, which provides a good initialization. Second, IQL implements a weighted behavioral cloning policy extraction step, which has previously been shown to allow for better online policy improvement compared to other types of offline constraints (Nair et al., 2020). To evaluate the finetuning capability of various RL algorithms, we first run offline RL on each

| Dataset | AWAC | CQL | IQL (Ours) |
|---|---|---|---|
| antmaze-umaze-v0 | 56.7 → 59.0 | 70.1 → **99.4** | **88.0** → **96.3** |
| antmaze-umaze-diverse-v0 | 49.3 → 49.0 | 31.1 → **99.4** | **67.0** → 49.0 |
| antmaze-medium-play-v0 | 0.0 → 0.0 | 23.0 → 0.0 | **69.0** → **89.2** |
| antmaze-medium-diverse-v0 | 0.7 → 0.3 | 23.0 → 32.3 | **71.8** → **91.4** |
| antmaze-large-play-v0 | 0.0 → 0.0 | 1.0 → 0.0 | **36.8** → **51.8** |
| antmaze-large-diverse-v0 | 1.0 → 0.0 | 1.0 → 0.0 | **42.2** → **59.8** |
| antmaze-v0 total | 107.7 → 108.3 | 151.5 → 231.1 | **374.8** → **437.5** |
| pen-binary-v0 | **44.6** → **70.3** | 31.2 → 9.9 | 37.4 → 60.7 |
| door-binary-v0 | **1.3** → **30.1** | 0.2 → 0.0 | 0.7 → **32.3** |
| relocate-binary-v0 | **0.8** → 2.7 | 0.1 → 0.0 | 0.0 → **31.0** |
| hand-v0 total | **46.7** → 103.1 | 31.5 → 9.9 | 38.1 → **124.0** |
| total | 154.4 → 211.4 | 182.8 → 241.0 | **412.9 → 561.5** |

Table 2: Online finetuning results showing the initial performance after offline RL, and performance after 1M steps of online RL. In all tasks, IQL is able to finetune to a significantly higher performance than the offline initialization, with final performance that is comparable to or better than the best of either AWAC or CQL on all tasks except pen-binary-v0.

dataset, then run 1M steps of online RL, and then report the final performance. We compare to AWAC (Nair et al., 2020), which has been proposed specifically for online finetuning, and CQL (Kumar et al., 2020), which showed the best performance among prior methods in our experiments in the previous section. Exact experimental details are provided in Appendix C. We use the challenging Ant Maze D4RL domains (Fu et al., 2020), as well as the high-dimensional dexterous manipulation environments from Rajeswaran et al. (2018), which Nair et al. (2020) propose to use to study online adaptation with AWAC. Results are shown in Table 5. On the Ant Maze domains, IQL significantly outperforms both prior methods after online finetuning. CQL attains the second best score, while AWAC performs comparatively worse due to much weaker offline initialization. On the dexterous hand tasks, IQL performs significantly better than AWAC on relocate-binary-v0, comparably on door-binary-v0, and slightly worse on pen-binary-v0, with the best overall score.

## 6 CONCLUSION

We presented implicit Q-Learning (IQL), a general algorithm for offline RL that completely avoids any queries to values of out-of-sample actions during training while still enabling multi-step dynamic programming. To our knowledge, this is the first method that combines both of these features. This has a number of important benefits. First, our algorithm is computationally efficient: we can perform 1M updates on one GTX1080 GPU in less than 20 minutes. Second, it is simple to implement, requiring only minor modifications over a standard SARSA-like TD algorithm, and performing policy extraction with a simple weighted behavioral cloning procedure resembling supervised learning. Finally, despite the simplicity and efficiency of this method, we show that it attains excellent performance across all of the tasks in the D4RL benchmark, matching the best prior methods on the MuJoCo locomotion tasks, and exceeding the state-of-the-art performance on the challenging ant maze environments, where multi-step dynamic programming is essential for good performance.

## ACKNOWLEDGEMENTS

We thank Dibya Ghosh and the anonymous reviewers for helpful comments on earlier drafts of the paper. This research was supported by the Office of Naval Research, C3.ai, and Intel, with compute support from Google.

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
