# OpenReview forum: "Offline Reinforcement Learning with Implicit Q-Learning"
_ICLR.cc/2022/Conference — ICLR 2022 Poster_

### Official Review · Reviewer_Sf4r · 2021-10-31

**Correctness:** 4
**Technical Novelty And Significance:** 3
**Empirical Novelty And Significance:** 2
**Recommendation:** 8
**Confidence:** 4

**Main Review:**

I believe that the main strength of this paper is the novel idea of using an expectile regression on dataset actions to learn a value function that is based on high-performing actions in the dataset. There have been many previous pieces of research on using quantile regression on RL, but most of them were to learn a distribution of value functions where the randomness comes from the environment, and the quantile regression is usually for the worst-case robustness. This paper, on the other hand, uses expectile regression on the value function where the randomness comes from the action and shows that it generalizes the Bellman expectation equation and Bellman optimality equation. As far as I know, this is the first suggestion of such an algorithm. The trick of using both $V$ and $Q$ functions to enable such learning seems to be very clever as well.

Theoretical analysis also backs up the algorithm well, and empirical performance is also presented well, showing its advantage over other algorithms. It is especially interesting that IQL works well on hard domains (antmaze, dexterous manipulation, etc.). IQL will be a simple and robust but high-performing tool for an offline RL. I also believe that a similar approach can be applied to standard RL algorithms, increasing their robustness.

It will be more interesting if the paper contains more analysis and experiments, such as the learned value functions for different $\tau$s and more state-of-the-art offline RL algorithms, but I am satisfied with the current form of the submission.

**Summary Of The Paper:**

This paper proposes an offline RL algorithm named IQL, which generalizes between Bellman expectation equation and Bellman optimality equation with expectile regression. Expectile regression assigns low weights for low-performing samples and assigns high weights for high-performing samples to learn the expectile. By performing $\tau$-expectile regression on the randomness of action in the dataset (from the sampling process of behavior policy), IQL is able to obtain a value function that is $\tau$ optimal. IQL is generalizing SARSA and Q-learning in the sense that SARSA corresponds to IQL with $\tau=0.5$ and Q-learning corresponds to IQL with $tau=1.0$. It offers very stable learning of a near-optimal value since it is only using in-distribution samples and the effective dataset size can be controlled through $\tau$. In the experiments, the paper has shown that IQL can obtain state-of-the-art performance.

**Summary Of The Review:**

The paper provides a good-performing robust offline algorithm based on a very simple but novel idea. I recommend acceptance of paper based on the novelty of the idea and the clean presentation.

---

### Official Review · Reviewer_SYQw · 2021-11-01

**Correctness:** 4
**Technical Novelty And Significance:** 3
**Empirical Novelty And Significance:** Not applicable
**Recommendation:** 5
**Confidence:** 4

**Main Review:**

The paper is well organized and easy to follow and read. The main contribution apart from prior work is the change in Q-function learning, which involves an additional value function learning and the expectile regression technique. After reading, I have the following question:
1.	The one-step and multi-step concepts come from Peng et al., 2019. I think in their case, the multi-step means iterative optimization between policy improvement and policy evaluation, so that it is the dynamic programming problem. In this paper, the multi-step of IQL is for learning Q function and value function?
2.	The learned optimal Q-function is for the learning of policy. Why advantage-weighted behavior cloning based on optimal Q-function could be better? I cannot see direct reasons here.
3.	From the proof, it is only when expectile tau goes to 1 in limit, so that the learned value function could be the optimal under the data. From the code, I observed that for MuJoCo tau=0.7, Adroit tau=0.7, and Ant-maze tau=0.9, which all do not goes limit to 1, like 0.99. I think there is a gap between the theory and the implementation.
4.	Given your learned optimal Q-function is pretty good and accurate, why not just optimize a parameterized policy (deterministic or Gaussian policy) based on the learned Q-function, for example maximizing E_(a,s) [Q(s,a)]. Why must use behavior cloning? BC is hard to exceed the best policy provided in the dataset.
5.	Is the value function really necessay in the framework? Do you try to remove the value function learning by just doing expectile regression over target Q values and Q values? The reason for asking this is that in continuous RL case, each state in the dataset is corresponding to one and only one action, which I think it is hard to train a good value function $V(s)$ with only one action $a$ by your eq(5).


**Summary Of The Paper:**

This paper proposes an algorithm In-Sample Q-Learning (IQL), which learns a Q- and value function under behavior policy and applies advantage-weighted behavior cloning for policy learning. In one sentence, the main idea of the paper is to learn an approximately optimal Q/value function for better behavior cloning. The expectile regression instead of MSE is applied to find optimal Q-function of behavior policy.

**Summary Of The Review:**

The idea is simple and straight forward, but more explanations are needed to provide strong insights.

---

> ### Author Response · Authors · 2021-11-11
> **Response**
>
>
> Thank you for your comments and feedback. We discuss clarifications below and would be happy to add more analysis to further elucidate why the algorithm works and justify the design decisions. Are there any additional suggestions that you would have to improve the explanation, or do you believe that our revisions address all of your concerns?
>
> We would also emphasize that our method attains excellent results across a range of evaluation scenarios, outperforming prior methods in terms of both final performance and runtime. We believe that simple and effective methods that attain state-of-the-art results are of interest to the ICLR community.
>
> **“The learned optimal Q-function is for the learning of policy. Why advantage-weighted behavior cloning based on optimal Q-function could be better? I cannot see direct reasons here.“**
>
> In our work, we focus only on approximating the optimal Q-function, and for policy extraction, we use Advantage-Weighted Behavioral Cloning for its simplicity.  In principle, other techniques can be used as well for policy extraction, this is not the main point of the paper. In addition to Advantage-Weighted behavioral cloning, one can also use techniques described in Onestep RL by Branfonbrener et al. 2021: BCQ and BRAC-style policy extraction. Due to strong empirical results demonstrated in Branfonbrener et al. 2021, we decided to use advantage-weighted behavioral cloning. We expanded the discussion in Section 4.3 to address this point.
>
> **“Given your learned optimal Q-function is pretty good and accurate, why not just optimize a parameterized policy (deterministic or Gaussian policy) based on the learned Q-function, for example maximizing E_(a,s) [Q(s,a)]. Why must use behavior cloning? BC is hard to exceed the best policy provided in the dataset.”**
>
> Q(s,a) learned with IQL is not defined for actions outside of the dataset.  For this reason, unconstrained maximization of Q(s,a) might lead to picking actions with incorrectly overestimated values. For this reason, we used advantage-weighted behavioral cloning that corresponds to maximizing Q(s,a) with respect to a KL-constraint that prevents getting out-of-distribution actions. We want to emphasize that we use not standard Behavioral Cloning but weighted behavioral cloning that has different properties. We updated Section 4.3 to clarify this point.
>
> **“Is the value function really necessay in the framework? Do you try to remove the value function learning by just doing expectile regression over target Q values and Q values? The reason for asking this is that in continuous RL case, each state in the dataset is corresponding to one and only one action, which I think it is hard to train a good value function V(s) with only one action a by your eq(5).”**
>
> We explain the necessity of learning a value function in Section 4.2. In particular, it’s important to learn a value function since otherwise maximization will be performed not only over $a’$ but also over $s’ \sim p(\cdot|s, a)$. We respectfully disagree with the statement that learning a good value function is difficult, as demonstrated by our experimental results. We also emphasize that different states are mapped into similar representations due to neural network generalization. Most methods for offline RL are based on this property of neural networks. For example, BCQ would sample exactly the same action, while CQL would have $-\infty$ values for all actions that do not appear in the dataset. We expanded the discussion in Section 4.2 for further clarification and updated Figure 3 to illustrate the effect of performing expectile regression for value learning.
>
> **“The one-step and multi-step concepts come from Peng et al., 2019. I think in their case, the multi-step means iterative optimization between policy improvement and policy evaluation, so that it is the dynamic programming problem. In this paper, the multi-step of IQL is for learning Q function and value function?”**
>
> Similarly to Peng et al., 2019, multi-step means iterative optimization between policy improvement and policy evaluation in our work. However, in our case, the policy improvement step is implicit and absorbed into learning V. In particular, V approximated the implicit argmax policy. Therefore, we updated Section 4.4 to address the question.
>
> **“From the proof, it is only when expectile tau goes to 1 in limit, so that the learned value function could be the optimal under the data. From the code, I observed that for MuJoCo tau=0.7, Adroit tau=0.7, and Ant-maze tau=0.9, which all do not goes limit to 1, like 0.99. I think there is a gap between the theory and the implementation.”**
>
> We agree with this observation, and we have already discussed it at the end of Section 4.4 (bottom of page 6). One of the reasons for using smaller values of tau is that it becomes numerically difficult to estimate larger expectiles of a distribution.

---

> > ### Author Response · Authors · 2021-11-18
> > **Update**
> >
> > Dear reviewer,
> >
> > Please let us know if our follow-up has addressed the issues raised in your review. We hope that our corrections, clarifications, and additional results address the concerns you've raised. We are happy to address any further concerns.

---

> > ### Comment · Reviewer_SYQw · 2021-11-25
> > **Some of my concerns are still there.**
> >
> > Thanks for your reponses. Some of my concerns are still there.
> >
> > 1. "The learned optimal Q-function is for the learning of policy. Why advantage-weighted behavior cloning based on optimal Q-function could be better? I cannot see direct reasons here." This is not asking for why you use advantage-weighted behavior cloning but for why using advantage-weighted behavior cloning with an approximate optimal Q-function could be better. For me, it is only that using approximate optimal Q function will filter the low-Q value state-actioin pairs for policy cloning.  I am not sure whether this is your intuition of using approximate optimal-Q function here. However, in this case, it will also suffers from the data sparsity issue if too many datapoints are fltered out.
> >
> > 2. "We want to emphasize that we use not standard Behavioral Cloning but weighted behavioral cloning that has different properties" I know you are using the advantage-weighted behavior cloning.
> >
> > 3. "For this reason, we used advantage-weighted behavioral cloning that corresponds to maximizing Q(s,a)" In your paper, the baseline methods already include BCQ and CQL, which are general offline RL methods that deal with the issue, "unconstrained maximization of Q(s,a) might lead to picking actions with incorrectly overestimated values". What I am asking is whether your proposed approximate optimal Q-function learning could benefits there methods? Then, it will constitute a direct and fair comparison between them.
> >
> > 4.  "In particular, it’s important to learn a value function since otherwise maximization will be performed not only over $a'$ but also over $s'$." The maximization is always on the expectation of $a'$, which is the same as value function. Most of the off-policy(SAC;DDPG;TD3) and offline RL algorithms(BCQ;CQL) don't include the value function learning. So I am not sure why learning of value function is important here for the framework.
> >
> > 5.  "We respectfully disagree with the statement that learning a good value function is difficult, as demonstrated by our experimental results." I didn't see the expriement results that show your learned value function is good. The final performance is influenced by many things, where you can not clearly see the contribution of components.
> >
> > 6. "the policy improvement step is implicit and absorbed into learning V." This confuses me. Your policy is only updated by advantage-weighted behavioral cloning. Why the policy improvement step is absorbed into learning V?
> >
> > 7. "We agree with this observation" If there is a gap between the theory and implementation, better demonstrate this clearly in the paper. In this sense, your method is not learning an approximate optimal Q-function, but a Q-function close to that?
> >
> > Currently, I will remain my score for this paper. Thanks again for your detailed responses!

---

> > > ### Author Response · Authors · 2021-11-25
> > > **Re (part 1): Some of my concerns are still there.**
> > >
> > >
> > > We believe there may be some misunderstanding about the paper. We would be happy to clarify this in the writing, though we are not permitted to edit the paper until the final version at this point.
> > >
> > > The main purpose of our paper is not to propose a better policy extraction method, but rather to develop a method for training a Q-function that effectively approximates an optimal Q-function, while avoiding the use of out-of-sample actions during training. This is accomplished by means of our expectile regression method. Once this process completes, it is then necessary to extract a policy from this Q-function. We perform this extraction using advantage-weighted regression in our implementation, but this step could be done in many different ways -- our use of advantage-weighted regression in this regard is not new, and it's not the contribution of the work. The contribution is the method for training Q-functions with expectile regression. We agree that other techniques could also be used for policy extraction (e.g., BCQ, CQL), and we would be happy to clarify this in the paper, but the specific choice of extraction method is not the main contribution of the work. We clarify the other specific points you raised below. Hopefully this helps to clarify the paper, please let us know if any other issues remain.
> > >
> > > > why using advantage-weighted behavior cloning with an approximate optimal Q-function could be better ... it will also suffers from the data sparsity issue if too many datapoints are fltered out
> > >
> > > We agree that using advantage-weighted regression for extracting the policy likely has shortcomings, and we would be happy to discuss these in the paper. We apologize for the confusion. We have already expanded the discussion of this issue in Section 4.3. We will expand the section further for the camera-ready version. However, the particular choice of AWR for extraction is not our contribution, and we could extract the policy with other methods too that do not suffer from this issue. We believe that the strong empirical results attained by our method suggest that on standard benchmark tasks, this sparsity issue is not a major problem, but we agree that it would be good to add discussion of it.
> > >
> > > > What I am asking is whether your proposed approximate optimal Q-function learning could benefits there methods? Then, it will constitute a direct and fair comparison between them.
> > >
> > > This is a good suggestion! It's quite likely that our method (IQL) could use a method like BCQ or CQL for policy extraction instead of AWR. We would be happy to add discussion of this, but we believe that our current comparison is fair: both BCQ and CQL are methods for learning good Q-functions via offline RL. Our method is also an algorithm for learning Q-functions, so it makes sense to compare our method to these prior approaches.
> > >
> > > > Most of the off-policy(SAC;DDPG;TD3) and offline RL algorithms(BCQ;CQL) don't include the value function learning. So I am not sure why learning of value function is important here for the framework.
> > >
> > > This is important due to the use of expectile regression. Using expectile regression in the Q-function update would induce optimism with respect to both a and s' (since target values are stochastic, with randomness due to both a and s'). This point is discussed in detail in Section 4.2 (see "However, this formulation has a significant drawback."). We will clarify this discussion further in the final paper. While the reason our method requires a separate value function is specific to our use of expectile regression, we would also note that some prior algorithms have indeed used a separate value function (see, e.g., Haarnoja et al. ["Soft Actor-Critic"](https://arxiv.org/pdf/1801.01290.pdf) Eq (5) and Eq (6), which discusses updating a separate V function).
> > >
> > > > I didn't see the expriement results that show your learned value function is good.
> > >
> > > You're right, we should not claim that the value function is accurate, because we do not test this (and, indeed, most RL papers do not test this). Currently, this statement does not appear in the paper.

---

> > > > ### Author Response · Authors · 2021-11-25
> > > > **Re (part 2): Some of my concerns are still there.**
> > > >
> > > >
> > > > > Your policy is only updated by advantage-weighted behavioral cloning. Why the policy improvement step is absorbed into learning V?
> > > >
> > > > The structure of our algorithm differs from that of standard actor-critic methods (see discussion in Section 4.3). Whereas actor-critic methods alternate between updating the actor and critic every step, our method first trains the critic to convergence via expectile regression, and then extracts an actor from it. When we say that there is an "implicit" policy improvement step, we mean that the expectile regression update trains the Q-function and value function to estimate the value of a policy that is better than the behavior policy, due to the maximization effect of expectile regression. We acknowledge that this phrasing is confusing though, so we will aim to revise it to clarify the distinction between the improvement in the expectile regression update (which has nothing to do with advantage-weighted regression), and the final AWR-based policy extraction step (which does not influence the critic in any way).
> > > >
> > > > > If there is a gap between the theory and implementation, better demonstrate this clearly in the paper. In this sense, your method is not learning an approximate optimal Q-function, but a Q-function close to that?
> > > >
> > > > Is there something in particular that you would like us to demonstrate? We did try to clarify in the paper that we use a value of tau less than 1 in practice in Section 4.4 (bottom of page 6). We believe that the good performance of the resulting method supports that this is a good choice. However, if there is anything else in this regard that you would like us to evaluate, we would be happy to add that!
> > > >
> > > > In regard to approximations: we would emphasize that all Offline RL learning methods do not learn an optimal function directly but rather estimate an optimal function plus some penalty. Please see the theoretical explanation in [CQL](https://arxiv.org/abs/2006.04779) where the authors discuss the gap between the CQL critic and optimal Q-values (Theorem 3.4).

---

### Official Review · Reviewer_DeQA · 2021-11-02

**Correctness:** 4
**Technical Novelty And Significance:** 3
**Empirical Novelty And Significance:** 2
**Recommendation:** 6
**Confidence:** 2

**Main Review:**

The paper overall is easy to follow and well written. I have two major concerns / questions:
1. For a task with continuous state space, most state may only have one action in the logging data, which means $\pi_\beta(a|s)$ maybe a Dirac distribution. If this is the case, $E_a Q(s,a)$ would be the same with $\max_{a\in A,  \pi_\beta(a|s)>0} Q(s,a)$. If this is the case, does it mean the SARSA-style policy evaluation and optimal Q-function are not different? Then what's the role of expectile regression under this case?
2. The results of IQL on Gym locomotion tasks are on-par or worse than those from CQL, it would be interesting to add some analysis of this. I found it a little bit counter-intuitive because IQL is more similar to behavior cloning than CQL, but underperforms on tasks with better quality data.

Minor:

1. Should it be $V_{\tau}(s) = E_{a\sim \pi_\beta (a|s)}[Q_{\tau}(s,a)]$ above Lemma 2?
2. It would be great to bold the best model on Table 1.

**Summary Of The Paper:**

This paper proposes in-sample Q-learning which composed of fitting value function with expectile regression, training Q-function and extracting the policy using advantage-weighted behavioral cloning. The method is validated on a set of D4RL examples and shows promising results on Ant Maze tasks.

**Summary Of The Review:**

In summary, I think the paper is well written and the idea is interesting. The empirical evaluation is significantly better than the existing baselines on challenging tasks like Ant Maze, and I thereby recommend accepting the paper.

---

> ### Author Response · Authors · 2021-11-11
> **Response**
>
> Thank you for your comments and feedback. We discuss clarifications below and would be happy to expand the discussion to clarify our paper further. We added a figure illustrating the difference between value functions estimated with SARSA and IQL (Figure 3, right). Also, we want to clarify that we don't think IQL is underperforming, but rather we think that performance on locomotion tasks is maxing out given that all recent methods perform similarly on these tasks. IQL matches (or slightly exceeds) the best prior method, CQL, because there is not much room to improve these tasks. This is also why we included ant maze and finetuning results. Please see expanded answers below. Does this reply address all of your concerns? Please let us know if any other outstanding issues remain, or what additional modifications or clarifications would be needed for you to raise your score.
>
> **“The results of IQL on Gym locomotion tasks are on-par or worse than those from CQL, it would be interesting to add some analysis of this. I found it a little bit counter-intuitive because IQL is more similar to behavior cloning than CQL, but underperforms on tasks with better quality data.“**
>
> IQL performs better or on par with CQL on medium and medium-replay datasets while the total score is slightly worse. However, we believe that the difference is not statistically significant since the difference is well within standard error. We focused on getting better performance specifically on these datasets since they are more challenging than medium-expert ones, which can be solved with behavioral cloning on top trajectories (see %BC in Table 1). Note that CQL is also similar to BC in some sense. If the TD loss is removed from CQL, it performs behavioral cloning with an EBM and then distills it into a Gaussian policy. We will be happy to add some analysis to clarify this, and we will appreciate any suggestions.
>
> **“For a task with continuous state space, most state may only have one action in the logging data, which means $\pi_\beta(a|s)$ maybe a Dirac distribution. If this is the case, $E_a [Q(s,a)]$ would be the same with $\max_ {a \in \mathcal{A}, \pi_{\beta}(a|s)>0} Q(s,a)$. If this is the case, does it mean the SARSA-style policy evaluation and optimal Q-function are not different? Then what's the role of expectile regression under this case?”**
>
> We agree with this observation. However, due to neural network generalization, different states are mapped into similar representations inducing an action distribution different from dirac. Most methods for offline RL are based on this property of neural networks. For example, BCQ and EmaQ would sample exactly the same action without generalization, while CQL would have $-\infty$ values for all actions that do not appear in the dataset. We thank the reviewer for raising this concern. We expanded the discussion in Section 4.2 and updated Figure 3 (right) with a plot that compares value functions learned with SARSA ($\tau=0.5$) and IQL ($\tau=0.7$, $0.9$).
>
> **“Should it be ... above Lemma 2?” “It would be great to bold the best model on Table 1.”**
>
> Thank you. We fixed it in the updated version of the submission.

---

> > ### Author Response · Authors · 2021-11-18
> > **Update**
> >
> > Dear reviewer,
> >
> > Please let us know if our follow-up has addressed the issues raised in your review. We hope that our corrections, clarifications, and additional results address the concerns you've raised. We are happy to address any further concerns.

---

> > ### Comment · Reviewer_DeQA · 2021-11-22
> > **thank you for the clarification**
> >
> > Thank you for the clarification. The response addressed my concerns in general, and I appreciate the additional experiments which makes the statement more convincing. Though the role of $\tau$ does not have a perfect match between theory and practice (as also brought up by reviewer SYQw), the additional learning curves show that larger $\tau$ ($\tau=0.9$) outperforms the ones with smaller $\tau$, which makes the choice reasonable.

---

> > > ### Author Response · Authors · 2021-11-23
> > > **Re: thank you for the clarification**
> > >
> > > Thank you for your response. Please let us know if there are other issues we can address that would enable you to increase your score for the paper.

---

### Official Review · Reviewer_eQvF · 2021-11-07

**Correctness:** 4
**Technical Novelty And Significance:** 2
**Empirical Novelty And Significance:** 2
**Recommendation:** 5
**Confidence:** 5

**Main Review:**

This paper studies how to avoid updating with out-of-sample or unseen actions during Q-updates which is one of the main challenges in offline RL as unseen actions can generate erroneous values for Q-functions and result in overestimation problems. While this paper looks at right place to address this issue, there are some concerns with novelty and design choices in this paper:

- [1] originally proposed to use SARSA-like TD update during policy evaluation step that doesn't require querying out-distribution actions either. While [1] applies their method only in the discrete-action space, there is significant overlap between this work and [1]. Importantly, the authors fail to acknowledge and discuss this important related work which also weakens their argument about novelty with regard to the in-sample policy evaluation step.

- There is lots of unknown about why expectile regression was selected for value function updates and it is not clear what is the motivation to go with expectile regression rather than mean regression. Particularly, what would happen if use mean regression, i.e. $L_V(\phi) = E_{(s,a)} [(Q_\hat{\theta}(s,a) - V_\phi(s))^2 ] $,  where it requires no $\tau$? How will performance change?  Can you run experiments by using MSE loss instead?

- One other idea is to use $V(s') = \frac{1}{N} \sum_{a' \sim \pi(.|s')} (Q_\hat{\theta}(s',a')] $ which is utilized in [4] ( see equation 1 in that paper). In addition, this work seems very similar to [4] as well. Can you discuss this and can you run experiments where $V(s)$ is estimated as mentioned?

- What happens if $V_\phi(s')$ is evaluated on s' in Eq 6 that is not trained on ( i.e s' is not shown during training of $V$ in Eq 5)?

- Looking at table 1, it seems your proposed method performs very similarly to others in locomotion-v2 and kitchen-v0, but marginally better on antmaze-v0. Since these results are very close, can you do significance testing or include their learning curves to provide a better picture about these results? Finally, can you also add BCQ results to Table 1?

- Can you explain how the learned policy is used during the deployment/test phase? Did you evaluate the learned policy with something like Algorithm 1 in [2]?

Related works:
 In the related work section, you only discussed offline methods where either policy or Q-function are regularized, not both. However, [3] shows that regularizing both policy and Q-function can be beneficial which is relevant to this paper as this paper also regularizes both Q and policy but implicitly. In addition, [2] also restricts the Q-function updates to [almost] state-action pairs that are in-distribution by learning a density function ( although there is still a chance for OOD action to be sampled by their density function). Both these works are missing and need to be discussed in the related work section.

[1] Regularized Behavior Value Estimation https://arxiv.org/abs/2103.09575

[2] EMaQ: Expected-Max Q-Learning Operator for Simple Yet Effective Offline and Online RL https://arxiv.org/abs/2007.11091

[3] Continuous Doubly Constrained Batch Reinforcement Learning https://arxiv.org/abs/2102.09225

[4] Keep Doing What Worked: Behavioral Modelling Priors for Offline Reinforcement Learning https://arxiv.org/abs/2002.08396



**Summary Of The Paper:**

This paper proposes a new offline RL algorithm that uses in-sample policy evaluation and advantage-weighted regression during policy improvement. Particularly, it utilizes sarsa-like TD to update Q-function and avoids the query value of unseen actions during training. To evaluate their proposed method, they used the D4RL benchmark to compare with previous offline RL methods. In addition, they show results for fine-tuning of learned offline-policies during online deployment.

**Summary Of The Review:**

While the general motivation of this paper about offline reinforcement learning is spot on, there are issues with this paper as mentioned above. Happy to hear authors' answers to raised issues.


**update after rebuttal**

Considering authors' responses and comments, I've increased my score.

---

> ### Author Response · Authors · 2021-11-11
> **Response**
>
> We thank the reviewer for the thorough and detailed comments. We believe there is a misunderstanding about our method, and we will try to clarify these points in the paper. We address specific points below, but first address what we think is the main misunderstanding. Our approach is not attempting to implement SARSA-based evaluation, in the sense that it is not attempting to estimate $Q^{\pi_\beta}$, although the form of the backup superficially resembles SARSA. Rather, it tries to estimate the optimal Q-function, and our main result is to show that this can be done without using out-of-sample actions via an expectile loss function. This is significantly different from prior methods that use a SARSA-style loss (i.e., [1], Onestep RL), because they are trying to estimate $Q^{\pi_\beta}$, typically via the MSE loss. As shown in our results in Tab. 1, Fig. 3, App. B (Tab. 3 and Fig. 4), and example in Fig. 2, our method performs significantly better than methods that only aim to estimate $Q^{\pi_\beta}$. We clarified the paper in Sec. 4 and 5.1 to better emphasize this contribution and avoid any future misunderstandings about the similarities between our method and SARSA.
>
> **“[1] originally proposed to use SARSA-like TD update. ”**
>
> As we discussed above, our method is significantly different from SARSA, and actually has a different goal (it is not trying to estimate $Q^{\pi_\beta}$). That said, we definitely missed the citation to [1], and we have now revised the paper to add a reference to [1] alongside Onestep RL to the related work and Sec. 5.1 where we discuss connections between our method and Onestep RL (Brandfonbrener et al., 2021) which utilizes an idea similar to [1] for continuous actions. Both Onestep RL and [1] fit the Q-function with SARSA and penalize out-of-distribution actions for policy extraction.
>
> **“Can you run experiments by using MSE instead?”**
>
> We apologize for not stating it more clearly but our paper already includes experiments with MSE (which is the expectile loss with tau = 0.5). Also, Onestep RL in Table 1 represents a version of our method with MSE used instead of expectile regression and a slightly different policy extraction strategy. In Figure 3, our method with tau=0.5 also corresponds to MSE. For MSE loss, our method gets a total score of 97.8 (Tab. 3, App. B). For tau=0.9, our method gets 378, indicating about a 3x improvement from using expectile loss over the MSE loss. We believe this indicates the benefits of our approach. We appreciate the comments, and we have revised the paper to make this connection with MSE more clear. In particular, we replaced tau=0.5 with tau=0.5 (MSE) in Fig. 1 and 3 to clarify this point and added complete results to App. B.
>
> **“this work seems very similar to [4]”**
>
> We appreciate the suggestion, we added preliminary ABM results, and we are working on improving our implementation. This will take a little while since there is no working public implementation that we are aware of, but we will aim to add this before the end of the rebuttal period. Note that the policy training objective we use is not a contribution of our paper, but is taken directly from prior work. Our main contribution is the expectile loss for Q-function training.
>
> **“Looking at table 1...”**
>
> Our method improves over recently proposed methods (e.g., TD3+BC, which was accepted for a spotlight presentation at NeurIPS, and Onestep RL (Brandfonbrener et al.), which was also accepted at NeurIPS) by a large margin. It attains an improvement on the locomotion tasks, and a 3x improvement on AntMaze. The closest prior method to ours in performance is CQL. However, on Ant Maze, which is by far the hardest of the tasks evaluated, we improve over CQL by 25% which is well in line with improvements in prior works. Note that TD3+BC did not improve on prior work in terms of performance (as can be seen in Table 1), but only in terms of simplicity. Our work both introduces a simpler method and improves over the prior work quantitatively. Furthermore, our method attains a very large improvement over CQL in runtime (4x faster), as well as in the finetuning experiments (2x improvement, Table 2). We believe that these improvements over prior work are significant by the standards used in prior offline RL papers. We added BCQ results and learning curves to App. B (Fig. 4 and 5).
>
> **“how the learned policy is used during the deployment/test phase?”**
>
> In deployment, we simply take the mean action from the policy extracted as described in Section 4.3. We do not evaluate sampled actions as in [2] or BCQ.
>
> **“What happens if  is evaluated on s' in Eq 6 that is not trained on?”**
>
> All s’ except the terminal transition s' of each trajectory appears in training. However, evaluating this s’ is significantly different from evaluating an action sampled from a training policy since s’ is in-distribution and does not suffer from the distribution shift similar to the policy actions in our methods.

---

> > ### Comment · Reviewer_eQvF · 2021-11-11
> > **thanks and clarfication**
> >
> > Thanks for your quick responses. I'll go through all your responses to all reviews and follow up with you. I have yet to read your responses carefully.
> >
> >
> > One quick clarification: I was simply asking to replace $V_\psi(s')$ in Eq 6 with $V(s') = \frac{1}{N} \sum Q(s',a')$ and re-run your algorithm and not necessarily reimplementing [4] . I understand that their code is not available.  Moreover, if you discuss how your method is different from [4] and highlight your contributions in compare to them, it'd be helpful.

---

> > > ### Author Response · Authors · 2021-11-12
> > > **Additional clarification**
> > >
> > > Thank you for the quick response. We believe that the paper already contains exactly this comparison, though it depends on how $a'$ is sampled in $Q(s',a')$.
> > >
> > > If $a'\sim\pi_\psi(\cdot|s')$ then we already have this result in the paper, it corresponds to AWAC in Table 1. Please see Section II and Algorithm 1, line 6 and 7, in AWAC: Accelerating Online Reinforcement Learning with Offline Datasets (https://arxiv.org/abs/2006.09359).
> > >
> > > If $a'\sim\pi_\beta(\cdot|s')$ then we have this result as well (Onestep RL, Table 1). Please see Eqn 2. in Offline RL Without Off-Policy Evaluation (https://arxiv.org/abs/2106.08909) which we call Onestep RL in our paper.
> > >
> > > Both methods differ only in critic learning schemes and share a similar policy extraction strategy with our method (see Algorithm 1, line 8 in AWAC and Eqn. 5 in Openstep RL).
> > >
> > > Please let us know if your suggestion is not covered by these methods, we will be happy to add it to the paper.

---

> > > > ### Author Response · Authors · 2021-11-14
> > > > **Additional results**
> > > >
> > > > We again appreciate the suggestions, and assuming that $V(s)$ is computed using several samples from the learned policy, we added comparisons to the value function estimated with our method (see Appendix E). Similarly to other baselines reported in Table 1, this estimator underperforms on AntMaze tasks. In particular,  we obtain 661.4 $\pm$ 93.7 in locomotion tasks (IQL gets 692.4 $\pm$ 52.1) and 153.9 $\pm$ 8.1 for Ant Maze tasks (IQL gets 378$\pm$49.9). For each state, we used 20 samples from the learned policy to estimate the value.
> > > >
> > > > We are happy to add more ablations and incorporate other suggestions into our submission. Please let us know if there are other issues we can address.

---

> > ### Comment · Reviewer_eQvF · 2021-11-19
> > **update after rebuttal**
> >
> > Thanks for your responses and efforts. You are right, $\tau=0.5$ and MSE are the same thing, I initially didn't notice $\tau=0.5$ in your experiments.
> >
> > While I am still on the fence about this paper, I've increased my score in light of new experiments and clarifications. That said, I'd suggest to the authors to have a discussion in the paper about the most critical part of their method as it is not clear to me what is the most critical part in their method: is it how V is estimated or how policy is updated? or both? Because Table 5 experiments show that estimating  $V(s')$ using  $\frac{1}{N} \sum Q(s',a')$ can be just as good as learning $V_\psi(s')$ (i.e. locomotion results are statistically similar, only ants results are different).
> >
> >
> > **More Questions**
> > - Can you confirm that the only difference between IQL and V(s) experiments in Table 5 is how V(s) is estimated and all other details are exactly the same?
> >
> > - Do you have results that compare IQL with IQL without AWA ( i.e. $E[ log \pi(a|s)]$) or IQL with $E_{a \sim \pi}[ Q (s, a)]$ to update the policy? (minor)
> >
> > - Does IQL need to have access to (s, a, r, s' , a') (i.e. trains $V_\psi(s)$ using (s',a')) or it only needs (s, a, r, s') and it uses (s, a) to train $V_\psi(s)$?
> >
> > Minor comment:
> > I'd rewrite Algorithm 1 in full details and add it to the appendix too. It is good that Algorithm 1 is in the main text, but not as informative as one wishes for, especially for implementation purposes.

---

> > > ### Author Response · Authors · 2021-11-19
> > > **Response to update after rebuttal**
> > >
> > > We thank the reviewer for raising the score and actively participating in the discussion. We really appreciate your time and effort.
> > >
> > > We want to emphasize that the main contribution is the use of expectile regression for training the value function which approximates the optimal value function in contrast to using mean squared error. The policy extraction step is standard, it is simply taken from prior work such as AWAC (Nair etal. 202), AWR (Peng etal. 2019), Onepstep RL (Brandfonbrener etal. 2021), and indeed it is quite likely that other policy extraction approaches could also be used. However, as demonstrated in BCQ, unconstrained policy extraction schemes fail in Offline RL; thus, we choose one of these constrained policy extraction schemes. We've clarified this point in Section 4.3 to avoid this misunderstanding in the future.
> > >
> > > **“Because Table 5 experiments show that estimating $V(s')$ using $\frac{1}{N} \sum Q(s',a')$ can be just as good as learning $V_\psi(s')$ (i.e. locomotion results are statistically similar, only ants results are different).”**
> > >
> > > Note that D4RL consists of several domains: Locomotion, Ant Maze, Kitchen, and Adroit.  And there is a big difference between the Ant Maze domains and the Locomotion tasks (see more in-depth discussion in the [D4RL paper](https://arxiv.org/abs/2004.07219)).  Moreover, many prior works, such as TD3+BC, Decision Transformers, AWAC, Onestep RL, focus on the locomotion tasks despite the fact that the locomotion tasks are simpler than the other domains. In particular, as shown in Table 5, even behavioral cloning of the top quantile of trajectories for the locomotion tasks already performs very well in these settings, hence there is simply not much room for improvement for any offline RL method, as evidenced by the fact that prior methods also behave similarly to %BC. Indeed, D4RL specifically proposes the ant maze tasks to address this challenge. And, in contrast, to these papers mentioned above, we evaluate our method on all domains. On the Ant Maze tasks specifically, there is a very large difference across the board between these methods, the MSE approach and our proposed expectile regression approach.
> > >
> > >
> > >
> > > **Responses to More Questions**
> > >
> > > **“Can you confirm that the only difference between IQL and V(s) experiments in Table 5 is how V(s) is estimated and all other details are exactly the same?”**
> > >
> > > We confirm.
> > >
> > > **“Do you have results that compare IQL with IQL without AWA ( i.e. ) or IQL with to update the policy? (minor)”**
> > >
> > > Please see our response above.
> > >
> > > **“Does IQL need to have access to (s, a, r, s', a') (i.e. trains  using (s', a')) or it only needs (s, a, r, s') and it uses (s, a) to train ?”**
> > >
> > > IQL doesn’t need access to (s, a, r, s’, a’) and is currently trained using only (s, a, r, s’) for Q and (s, a) for V. This is standard practice for fitting V; please, see the original [soft actor-critic paper](https://arxiv.org/abs/1801.01290) for reference. However, an alternative version can be considered that uses  (s, a, r, s’, a’)  instead.

---

> > > ### Author Response · Authors · 2021-11-25
> > > **Re: update after rebuttal**
> > >
> > > Dear Reviewer,
> > >
> > > Thank you for continuing the discussion. We would appreciate a response from you as to whether our clarifications address all of your points and your question about what is the most important part of the paper. We believe we've responded to these issues in full, but if there is something that is still missing, let us know and we would be happy to add it. If we've addressed your concerns fully, however, we would appreciate it if you would revise your score.

---

### Author Response · Authors · 2021-11-11
**Revision summary**

We would like to thank the reviewers for their detailed comments. We respond to the individual reviews below. We’ve also updated the paper with a number of modifications to address reviewer suggestions and concerns. Summary of updates:
1) We added a citation to R-BVE alongside the other “one step” method (Onestep RL, Brandfonbrener et al. 2021);
2) We added BCQ results to Table 1;
3) We bolded results in [0.95 * best, best] in Table 1;
4) We added standard deviations for our method and for prior methods for which this information was available;
5) We updated the legends for Figure 1 (left) and Figure 3 (left) to clarify that tau=0.5 corresponds to MSE (SARSA-style policy evaluation);
6) We added the curves for learned V values for different values of tau with and without Clipped Double Q-Learning (Figure 3, right);
7) We expanded the discussion in Sections 4, 4.2, 4.3, 4.4, 5.1 and 5.2 to address concerns raised by the reviewers;
8) We added full comparisons to MSE on Ant Maze tasks to Appendix B (Figure 5 and Table 3);
9) We added learning curves for locomotion and Ant Maze tasks to Appendix B (Figure 4 and 5);
10) We added additional references to the related work section:
- Regularized Behavior Value Estimation https://arxiv.org/abs/2103.09575
- EMaQ: Expected-Max Q-Learning Operator for Simple Yet Effective Offline and Online RL https://arxiv.org/abs/2007.11091
- Continuous Doubly Constrained Batch Reinforcement Learning https://arxiv.org/abs/2102.09225
- Keep Doing What Worked: Behavioral Modelling Priors for Offline Reinforcement Learning https://arxiv.org/abs/2002.08396

---

### Decision · Program_Chairs · 2022-01-20

**Decision:**

Accept (Poster)

**Comment:**

This paper proposes a new paradigm --- called in-sample Q learning --- to tackle offline reinforcement learning. Based on the novel idea of using expectile regression, the proposed algorithm enjoys stable performance by focusing on in-sample actions and avoiding querying the values of unseen actions. The empirical performance of the proposed algorithm is appealing, outperforming existing baselines on several tasks. The paper is also well written.